# Double-Negative T-Cell Reaction in a Case of *Listeria* Meningitis

**DOI:** 10.3390/ijerph18126486

**Published:** 2021-06-16

**Authors:** Asad Ullah, G. Taylor Patterson, Samantha N. Mattox, Thomas Cotter, Nikhil G. Patel, Natasha M. Savage

**Affiliations:** 1Department of Pathology, Medical College of Georgia, Augusta University, Augusta, GA 30912, USA; aullah@augusta.edu (A.U.); smattox@augusta.edu (S.N.M.); tcotter@augusta.edu (T.C.); NPATEL4@augusta.edu (N.G.P.); 2Medical College of Georgia, Augusta University, Augusta, GA 30912, USA; GPATTERSON1@augusta.edu

**Keywords:** listeria, altered mental status, pleocytosis

## Abstract

Gamma delta T-cells are commonly found in response to *Listeria monocytogenes* infection in mice, whereas this same immunological response has only been reported a few times in vivo in humans. Moreover, gamma delta T-cell response in cerebral spinal fluid samples in conjunction with *Listeria* meningitis has never been described in medical literature to date. Thus, we describe a 64-year-old male who presented with altered mental status, fever, and neck stiffness. After lumbar puncture revealed elevated glucose, protein, lactate dehydrogenase, and white blood cell count, further cytologic analysis was indicated. The CSF showed a markedly hypercellular sample with a lymphocytic pleocytosis, including some enlarged forms with irregular nuclear contours, and rare macrophage containing intracytoplasmic bacteria. Lymphocyte immunophenotyping was performed via flow cytometric analysis, which ultimately revealed a prominent CD4/CD8 negative T-cell population, suggestive of a gamma delta T-cell population. Thus, an initial suspicion of malignancy was considered but was ruled out due to the absence of mass lesion on imaging and overall features including heterogenous lymphocyte morphology. Shortly after, gram stain and cultures were obtained revealing *Listeria monocytogenes*. Unfortunately, the patient rapidly succumbed to disease following the diagnosis of *Listeria* meningitis. Studies suggest that gamma delta T-cells are activated by the protein components of *Listeria* and thus have been found to be an important mediator of resistance to *Listeria* infection. Studies have also discovered that the level of activation for these T-cells appears to be tissue specific and dose dependent, with most cases occurring within visceral organs. Hence, we herein present the first case of gamma delta T-cell activation due to *Listeria monocytogenes* within the cerebral spinal fluid of a human patient.

## 1. Introduction

*Listeria monocytogenes* are a group of gram-positive, rod-shaped bacteria often found in a variety of ecological niches due to their ability to act as a facultative anaerobe and motility [1,2]. The pathogen has been shown to cause a wide variety of adverse pathologic states including febrile gastroenteritis, septicemia, meningoencephalitis, and meningitis [3,4,5,6]. Infections from the bacteria are seen most frequently within persons in an immunocompromised state, pregnant women, and individuals at extreme ages (neonates and older adults). *L. monocytogenes* are often thought of as a foodborne pathogen commonly associated with consumption of un-pasteurized cheeses and cold deli meats [3,7]. Depending upon the route of inoculation, the pathogenesis and onset of symptoms can vary drastically. For instance, the mean incubation period from the time of infection to development of gastroenteritis is around 24 h with typical clinical features of fever, osmotic diarrhea, nausea, vomiting, headache, arthralgias, and myalgias. However, more serious invasive disease states, including septicemia and meningitis, have much longer incubation periods lasting around 11 days post inoculation [8]. Individuals presenting with *Listeria* related invasive disease often have the constitutional symptoms of fever and chills, but also have a much greater risk of developing central nervous system manifestations [8].

The pore-forming cytolysin, listeriolysin O (LLO), is the main virulence factor *L. monocytogenes* uses in order to act as a facultative intracellular pathogen during the course of an infection [9]. Its ability to easily replicate inside mononuclear phagocytes and T-cells make these immune cells particularly important in the anti-listerial resistance of an organism [1,10]. More specifically, gamma delta T-cells have been shown to increase dramatically in the initial phase of *Listeria* infection and are thus key in reducing the negative consequences associated with the infection.

Gamma delta T-cells were first discovered in the early 1990s, and therefore not every function has been fully elucidated. However, one major beneficial role these immune cells have is the ability to rapidly increase production of cytokines, resulting in an expansion of the immune response when a pathogen is encountered. The production of interferon gamma (IFN-γ) is among the most important cytokines produced in response to a *L. monocytogenes* infection [11,12]. In addition, and equally important, gamma delta T-cells down regulate the inflammatory response once the infection has resolved [11]. In general, gamma delta T-cell proliferation in response to a *Listeria* infection has been found to reside within the blood and gastrointestinal tract via mouse models. However, the case presented within this manuscript describes a unique situation in which a drastic increase in gamma delta T-cells was found within the cerebrospinal fluid (CSF) of a patient who presented with bacterial meningitis caused by *Listeria monocytogenes* infection.

## 2. Case Report

We present a 64-year-old male patient who arrived at the emergency department with altered mental status, fever, and severe neck stiffness. Notable past medical history of the patient included hepatitis C, cirrhosis, and end stage renal disease. During physical examination, the presence of severe nuchal rigidity was noted and thus a lumbar puncture (LP) was indicated. Upon performing the LP, a normal opening pressure was exhibited; however, analysis of the CSF showed significantly elevated glucose, protein, lactate dehydrogenase, and white blood cell count levels. Further cytologic analysis revealed a markedly hypercellular sample with a lymphocytic pleocytosis, including some enlarged potentially reactive forms displaying irregular nuclear contours. In addition, rare macrophages containing intracytoplasmic bacteria were seen (Figure 1). Flow cytometric immunophenotyping revealed a prominent CD4/CD8 negative T-cell population with decreased CD5 expression, suggestive of gamma delta T-cells (Figure 2). A suspicion of malignancy was initially considered, but CT imaging showed the absence of any notable mass lesion, effectively ruling this out from the differential diagnosis. Moreover, the heterogenous appearance of the lymphocytes favored a reactive process. Soon thereafter, gram stain and cultures recovered from the CSF revealed the bacteria *Listeria monocytogenes (*Figure 3). This finding ultimately confirmed the diagnosis of *Listeria* meningitis. However, despite extensive care and intervention, the patient’s condition rapidly deteriorated. Within several hours of diagnosis, the patient had unfortunately succumbed to disease.

## 3. Discussion

*Listeria monocytogenes* is a facultative intracellular gram-positive bacillus that is typically found in contaminated food [7]. It is the third most common cause of bacterial meningitis, especially in those who are immunocompromised or over the age of 50 [13]. Previous case reports have demonstrated that almost all patients with *Listeria* meningitis will present with at least 2 of the 4 classic symptoms of headache, fever, neck stiffness, and altered mental status [7]. However, differentiating *Listeria* meningitis from other bacterial meningitis infections can be challenging due to similar presenting symptoms and laboratory results. Thus, in order to definitively diagnosis a patient with *Listeria* meningitis, CSF examination must be performed. Several key characteristic changes are often witnessed within the CSF analysis, including lymphocytic pleocytosis and increased protein concentrations [14,15]. It should be noted that CSF Gram staining usually yields negative results due to the pathogen not taking up Gram stains well in clinical settings [16]. Therefore CSF cultures are the preferred method of diagnosis, with several case reviews revealing a CSF positivity rate around 75% [8,14].

Treatment options for *Listeria* meningitis include intravenous ampicillin combined with an aminoglycoside, or IV penicillin; the former being first line treatment due to the great efficacy seen with duel IV antibiotic therapy compared to monotherapy [3]. Mortality due to *Listeria* meningitis is relatively high, approximately 32.3%, and over 55% of survivors have neurologic sequelae. Thus, it is imperative to quickly and accurately diagnosis the patient with meningitis and begin delivering empiric antibiotic therapy in order to limit CNS damage and death [17].

Several studies have been conducted looking at the pathogenesis and immune response of *Listeria* infections in mouse models [2,11,12,18]. Most of these studies reveal that initial increases in gamma delta T-cells are imperative to an effective immune response to the bacteria. Gamma delta T-cells are immunogenic cells that are derived from the thymus and have a rapid, innate-like response to specific organisms at specific anatomical sites [18]. It has been determined that protein component listeriolysin O (LLO) or metabolites such as (E)-4-hydroxy-3-methyl-but-2-enyl pyrophosphate (HMBPP) and isopentenyl pyrophosphate (IPP) can act as specific ligand activators for gamma delta T-cells [2]. In addition, Belles et al. describes the level of activation of these T-cells to be tissue specific and dose dependent [18]. Studies performed on the liver, spleen, and intestines of mice revealed variable response times, but an overall significant increase in gamma delta T-cells. To date, very few in vivo human studies have illustrated a substantial gamma delta T-cell proliferation in response to a *Listeria* infection. However, one human study has been conducted involving peripheral blood mononuclear cells (PBMC) in vitro, with data indicating a similar immunologic response as displayed in mice in response to *Listeria* infection [19,20]. Thus, the case report described above is extremely unique with very few cases in medical literature illustrating this type of physiological response in vivo, let alone within the CSF.

## 4. Conclusions

Gamma delta T-cell activation and proliferation in *Listeria* infections is most prominent in the visceral organs of mice. However, in humans, only few in vivo cases have incited a similar response. This case report demonstrates one of the first known illustrations of the initial proliferation of double-negative T-cells, presumed gamma delta T-cells, within the cerebrospinal fluid of a human in response to an infection from *Listeria monocytogenes*. Identification of this unique immunologic response was made possible by histologic examination and flow cytometry analysis. It is our hope that this case report will aid in further investigation to define the role of gamma delta T-cells in humans.

## Figures and Tables

**Figure 1 ijerph-18-06486-f001:**
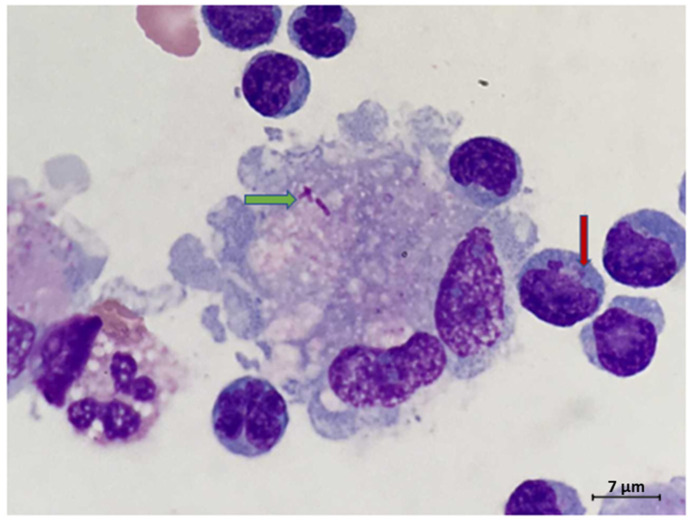
Rare macrophage with intracytoplasmic bacteria (green arrow). Note the lymphocytic pleocytosis in the background, including lymphocytes with irregular nuclear contours. Some lymphocytes contained intracytoplasmic granules (red arrow) (Wright Giemsa CSF sample, original magnification 1000×).

**Figure 2 ijerph-18-06486-f002:**
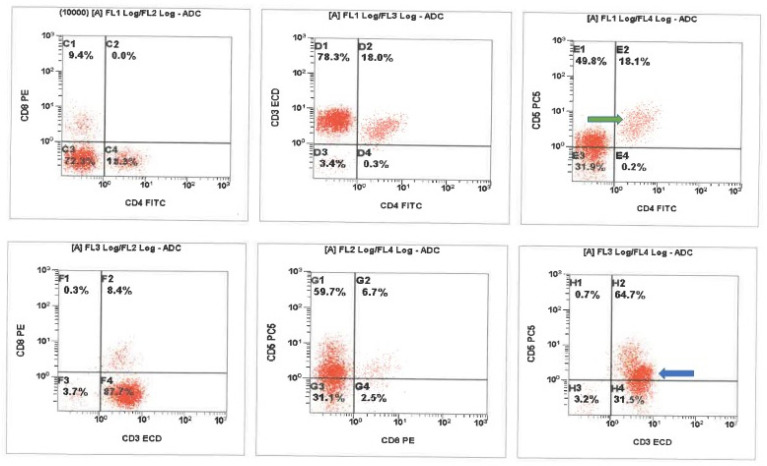
Cerebrospinal fluid (CSF) flow cytometric immunophenotyping revealed a background population of presumed alpha-beta T-cells with typical CD5 expression (green arrow) and a prominent CD4/CD8 negative T-cell population with decreased CD5 expression. The presumed gamma delta T-cells (blue arrow) show brighter CD3 expression as compared to the background presumed alpha-beta population. ADC-analogue to digital converter, FITC-fluorescein isothiocyanate, ECD-energy coupled dye (phycoerythrin-Texas Red conjugate).

**Figure 3 ijerph-18-06486-f003:**
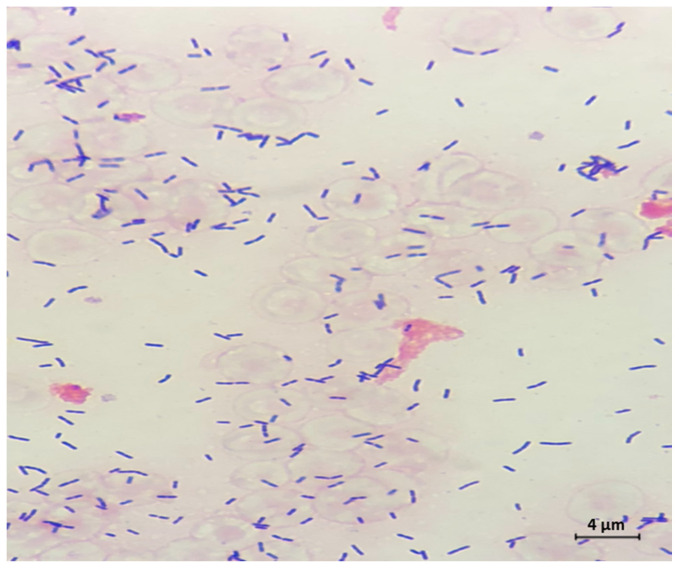
Cerebrospinal fluid (CSF) Gram stain revealed Gram positive rods, and culture later confirmed *Listeria monocytogenes* (original magnification 1000×).

## Data Availability

No new data were created or analyzed in this study. Data sharing is not applicable to this article.

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
