# Peer review of "Double-Negative T-Cell Reaction in a Case of Listeria Meningitis"

_ijerph, 2021, doi:10.3390/ijerph18126486_

Round 1

Reviewer 1 Report

Uhlah et al. reported on a case of Listeria monocytogenes with presence of gdT-cell in CSF, representing the first case describing such important aspect of this severe disease.  The information is, therefore, very relevant for clinical practice in other potential patients. However, there are a few minor comments to improve the manuscript.

  1. They should include a scale bar in Fig 1 and 3. 
  2. They should include more information in all figure legends to make easier the reading. For example, although in the text is very clear, they do not mention in the figure legend what type of sample (CSF) is being evaluated.
  3. Authors might address if they looked at the gdT-cell residing within the blood of the patient and if not, why?
  4. Figure 2 is too simple and even confusing- they are looking for negative CD4/CD8 population, so maybe it would be better to gate CD3 and then plot CD4 vs. CD8. Here we could easily distinguish those double negative cells. The way that they are showing Figure 2, we don’t know how many of the CD3+CD4- cells are CD8 (left), and the same (how many are CD4) at the  right. Could they include a figure more detailed? What is D and what is F? What is the key population? They might want to include some information about the processing of the sample for flow cytometry.
  5. Maybe is not necessary for this manuscript, but it would be great to have in Figure 2 the result of a patient without Listeria as control. I understand the difficulty of this comment but just to be considered in future works.

Overall, this is an elegant study worth of publishing, with a valid information for the audience.

Author Response

  1. They should include a scale bar in Fig 1 and 3.  

      Answer: Changes made as per your recommendation.

  1. They should include more information in all figure legends to make easier the reading. For example, although in the text is very clear, they do not mention in the figure legend what type of sample (CSF) is being evaluated. 

Answer: Changes made as per your recommendation. Figure 2 is replaced with more detailed information on flow studies.

  1. Authors might address if they looked at the gdT-cell residing within the blood of the patient and if not, why? 

Answer: This was not done as the patient rapidly deteriorated. 

  1. Figure 2 is too simple and even confusing- they are looking for negative CD4/CD8 population, so maybe it would be better to gate CD3 and then plot CD4 vs. CD8. Here we could easily distinguish those double negative cells. The way that they are showing Figure 2, we don’t know how many of the CD3+CD4- cells are CD8 (left), and the same (how many are CD4) at the right. Could they include a figure more detailed? What is D and what is F? What is the key population? They might want to include some information about the processing of the sample for flow cytometry. 

Answer: Changes made as per your recommendation. Figure 2 completely changed with more information about flow cytometric studies of the case.   

  1. Maybe is not necessary for this manuscript, but it would be great to have in Figure 2 the result of a patient without Listeria as control. I understand the difficulty of this comment but just to be considered in future works. 

Answer: Figure 2 is changed with more information about flow cytometry study of our case.

Reviewer 2 Report

In the submitted manuscript, Ullah et.al reported a case of Listeria meningitis. The authors documented clinical manifestations of a patient with fever and severe neck stiffness, collected CSF sample, and found in lymphocytic pleocytosis. The authors performed flow cytometry analysis and identified a CD3+CD4−CD8− T cell population in patient CSF. Bacteria culture later confirmed presence of Listeria monocytogenes in CSF. This is a very interesting case report. Data from this study will provide valuable information to the people in the field who are interested in this topic.

Here are some suggestions for authors to consider.

1. CD3+CD4−CD8− T cells (double-negative T cells; DNTs) population is not equivalent to gamma-delta T cell. T cell receptor repertoire analysis is needed to confirm the gamma-delta T cell. I would suggest authors to change the description to double-negative T cells in title and conclusion.

2. It is hard to understand figure 2. Gating strategies used in figure 2 is not clearly described. Providing more detail information about how authors performed FACS analysis will be helpful to understand the figure.

Author Response

  1. CD3+CD4−CD8− T cells (double-negative T cells; DNTs) population is not equivalent to gamma-delta T cell. T cell receptor repertoire analysis is needed to confirm the gamma-delta T cell. I would suggest authors to change the description to double-negative T cells in title and conclusion.

Answer:  These are presumed gamma delta T-cells based on overall immunophenotype. We did not have gamma delta antibody at that time. Changed to double negative T-cells, presumed gamma delta T-cells.

2. It is hard to understand figure 2. Gating strategies used in figure 2 is not clearly described. Providing more detail information about how authors performed FACS analysis will be helpful to understand the figure. 

Answer: Changes made as per your recommendation. Figure 2 completely changed with more information about flow cytometric studies of the case.   

Reviewer 3 Report

The manuscript of A. Ullah et al. "Atypical Gamma-Delta (γδ) T-cell Reaction in a Case of Listeria Meningitis" is devoted to the rare clinical case report of Listeria meningitis. The manuscript is well written. In this form, it fulfills the task of a research article.

I only have a few minor comments and suggestions:

Figure 1: I suggest using arrows to point out lymphocytes, intracytoplasmic granules, etc.
Figure 2: I suggest using a high-resolution image.
Line 137: Italisize "in vivo".
Sections Author Contributions, Funding, Acknowledgments, Conflicts of Interest are absent.
Line 153: Omit "Published 2020 Aug 30".
Line 168: Omit "Published 2020 Jun 30".

Author Response

  • Figure 1: I suggest using arrows to point out lymphocytes, intracytoplasmic granules, etc.

Answer: Changes made as per your recommendation.

  • Figure 2: I suggest using a high-resolution image.

Answer: Changes made as per your recommendation. Image is completely changed with more information and high resolution.

             3- Line 137: Italisize "in vivo".

Answer: Changes made as per your recommendation.

             4- Sections Author Contributions, Funding, Acknowledgments, Conflicts of Interest are absent.

Answer: Changes made as per your recommendation.

             5- Line 153: Omit "Published 2020 Aug 30".

Answer: Changes made as per your recommendation.

            6- Line 168: Omit "Published 2020 Jun 30" 

Answer: Changes made as per your recommendation.

Round 2

Reviewer 2 Report

The manuscript has been sufficiently improved.

This manuscript is a resubmission of an earlier submission. The following is a list of the peer review reports and author responses from that submission.